# Miniaturization of a Koch-Type Fractal Antenna for Wi-Fi Applications

**Dmitrii Tumakov** [1,*] , **Dmitry Chikrin** [2] **and Petr Kokunin** [2]

[1] Institute of Computer Mathematics and Information Technologies, Kazan Federal University; 18 Kremlevskaya Str., 420008 Kazan, Russia
[2] Institute of Physics, Kazan Federal University; 18 Kremlevskaya Str., 420008 Kazan, Russia; dmitry.kfu@gmail.com (D.C.); PAKokunin@kpfu.ru (P.K.)
* Correspondence: dtumakov@kpfu.ru

**Abstract:** Koch-type wire dipole antennas are considered herein. In the case of a first-order prefractal, such antennas differ from a Koch-type dipole by the position of the central vertex of the dipole arm. Earlier, we investigated the dependence of the base frequency for different antenna scales for an arm in the form of a first-order prefractal. In this paper, dipoles for second-order prefractals are considered. The dependence of the base frequency and the reflection coefficient on the dipole wire length and scale is analyzed. It is shown that it is possible to distinguish a family of antennas operating at a given (identical) base frequency. The same length of a Koch-type curve can be obtained with different coordinates of the central vertex. This allows for obtaining numerous antennas with various scales and geometries of the arm. An algorithm for obtaining small antennas for Wi-Fi applications is proposed. Two antennas were obtained: an antenna with the smallest linear dimensions and a minimum antenna for a given reflection coefficient.

**Keywords:** Koch-type antenna; fractal antenna; antenna miniaturization; Wi-Fi applications

## 1. Introduction

Wire antennas are widely used in modern telecommunication systems [1]. Despite the fact that the simplest wire half-wave dipole antennas are well explored [2], antennas with complex geometry are a separate subject for investigation [3]. The latter antennas are by far the most promising devices [4,5]. By complicating the geometry, it is possible both to minimize the dimensions of the antenna itself and to improve its electrodynamic characteristics [6,7].

In practice, various forms of broken balanced dipoles are used [1,8,9]. However, the most common way to minimize or improve the properties of the antennas is through their fractalization [10–16]. The most studied fractal antenna is a dipole constructed on the basis of the Koch fractal. A sufficient number of works have been devoted to the study and analysis of the base characteristics of both the Koch dipole and monopole [17,18], as well as its various modifications [19–21]. One of these modifications is the Koch-type fractal [22]. In addition, some other fractal structures are also used, including the Minkowski curve [23], Spidron fractal [24], crinkle curve [25], Peano fractal [26], and many other curves. Hybrid structures representing combinations of the above are also popular [27]. For designing antennas, in practice, both two-dimensional and three-dimensional structures are applied [28,29].

For the simplest half-wave dipole, especially in the case of the base frequency, the interconnections between its various parameters are well known [30]. The relationship between the base frequency and the reflector geometry was also investigated by the authors for Koch-type dipoles [22,31]. Such a relationship can be used for the designing of antennas [32]. We previously used this approach to model a Koch-type dipole of the first iteration for Wi-Fi applications [33].

In this paper, a Koch-type miniature wire antenna is designed using regression analysis. A second-order prefractal is chosen for the arm geometry. This choice is due to the fact that, because of the small linear dimensions of the dipole, it is technically very difficult to construct antennas with a complex geometry (higher-order prefractals). It appears to be possible only by means of significant reduction in the diameter of the wire.

## 2. Koch-Type Prefractals

A Koch-type fractal is not a classical fractal; it received its name due to its similarity to the Koch fractal [18]. The first iterations of these fractals differ only in the position of the central vertices (see Figure 1). When constructing iterations of Koch-type fractals, a fractal interpolation algorithm is used [34,35]. A description of this algorithm is given in detail below.

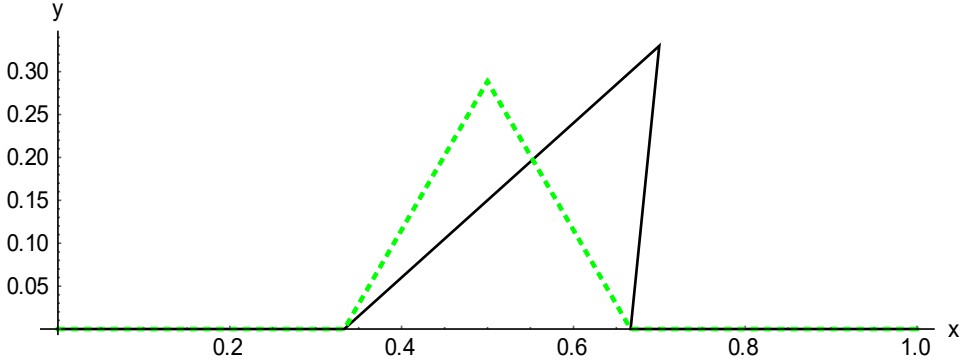

**Figure 1.** First iteration. Black solid line for the Koch-type fractal, green dashed line for the classical Koch fractal.

Let

$$K_1 = \{(t_i, x_i, y_i) \in [0,\ 1] \times R \times R \,\big|\, 0 = t_0 < t_1 < \ldots < t_n = 1\} \tag{1}$$

be some interpolation points for $i = 1..n$. For the Koch-type fractal, the number $n = 4$ is chosen. The values $x_0 = 0$, $x_1 = 1/3$, $x_3 = 2/3$, $x_4 = 1$ and $y_0 = y_1 = y_3 = y_4 = 0$ at arbitrary $x_2$ and $y_2 > 0$ form interpolation points for a family of Koch-type curves. Also, for antenna applications, the restriction $0 < x_2 < x_4$ is imposed. Note that in the case $x_2 = 1/2$, $y_2 = 1/\sqrt{12}$, the first iteration of the classical Koch curve is obtained.

For all $i$, the affine transformation is introduced according to the following rule:

$$A_i : R^3 \to R^3, \quad A_i \begin{pmatrix} t_i \\ x_i \\ y_i \end{pmatrix} := \begin{pmatrix} a_i & 0 & 0 \\ c_{i1} & D_{i1} & D_{i2} \\ c_{i2} & D_{i3} & D_{i4} \end{pmatrix} \begin{pmatrix} t_i \\ x_i \\ y_i \end{pmatrix} + \begin{pmatrix} e_i \\ d_{i1} \\ d_{i2} \end{pmatrix}. \tag{2}$$

Here, the matrices $D_{ij}$ have the form

$$D_{i1} = D_{i4} = \begin{pmatrix} 0.333 & 0 \\ 0 & 0.333 \end{pmatrix}, \quad D_{i2} = \begin{pmatrix} 0.167 & -0.289 \\ 0.289 & 0.167 \end{pmatrix}, \quad D_{i3} = \begin{pmatrix} 0.167 & 0.289 \\ -0.289 & 0.167 \end{pmatrix}. \tag{3}$$

Next, we require that all $i$ satisfy the following conditions:

$$A_i \begin{pmatrix} t_0 \\ x_0 \\ y_0 \end{pmatrix} = \begin{pmatrix} t_{i-1} \\ x_{i-1} \\ y_{i-1} \end{pmatrix}, \qquad A_i \begin{pmatrix} t_4 \\ x_4 \\ y_4 \end{pmatrix} = \begin{pmatrix} t_4 \\ x_4 \\ y_4 \end{pmatrix}. \tag{4}$$

Then, the following is obtained:

$$a_i = t_i - t_{i-1}, \qquad e_i = t_{i-1}, \tag{5}$$

$$\begin{pmatrix} c_{i1} \\ c_{i2} \end{pmatrix} = \begin{pmatrix} x_i - x_{i-1} \\ y_i - y_{i-1} \end{pmatrix} - \begin{pmatrix} D_{i1} & D_{i2} \\ D_{i3} & D_{i4} \end{pmatrix} \begin{pmatrix} x_4 - x_0 \\ y_4 - y_0 \end{pmatrix}, \qquad \begin{pmatrix} f_{i1} \\ f_{i2} \end{pmatrix} = \begin{pmatrix} x_{i-1} \\ y_{i-1} \end{pmatrix} - \begin{pmatrix} D_{i1} & D_{i2} \\ D_{i3} & D_{i4} \end{pmatrix} \begin{pmatrix} x_0 \\ y_0 \end{pmatrix}. \tag{6}$$

With this definition of the operators $A_i$, the straight line segment connecting the points $x_0$ and $x_4$ merges into a polyline, connecting the interpolation points in a consecutive manner.

Thus, a set of points for the second and third iterations is obtained as follows:

$$K_2 = \bigcup_{i=1}^{n} A_i(K_1), \quad K_3 = \bigcup_{i=1}^{n} A_i(K_2).$$

The graph of the obtained Koch-type prefractals is presented in Figure 2.

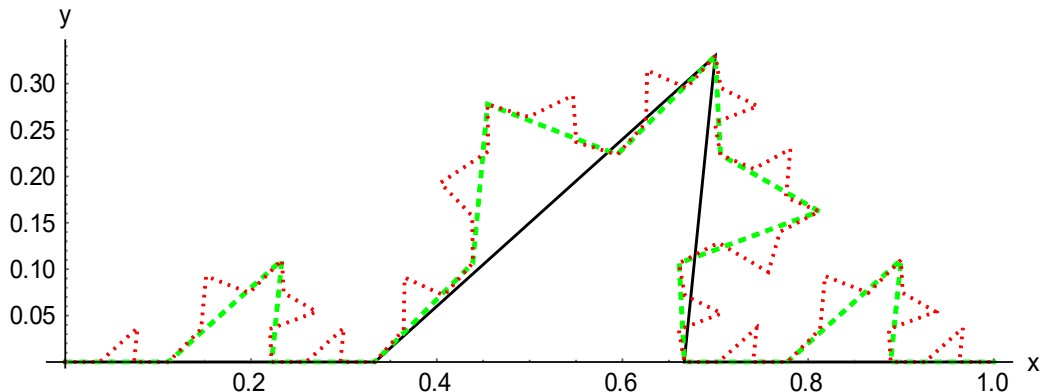

**Figure 2.** First (black solid line), second (green dashed), and third (red dotted) iterations of the Koch-type fractal. Coordinates of the central vertex: $x_2 = 0.7$, $y_2 = 0.33$.

Note that the transformation (2) is an affine transformation, not a dilatation; therefore, the curves shift from their original lines (Figure 2). In other words, the sets are self-affine sets, and the formulas are valid for the dimension $S$ of self-similar sets ([36], p. 130–132):

$$\sum_{i=1}^{n} r_i^S = 1, \tag{7}$$

$r_i = \sqrt{(y_i - y_{i-1})^2 + (x_i - x_{i-1})^2}$ is not valid in this case. The values of $S$ obtained via Equation (7) can serve only as an upper estimate of the fractal dimension of self-affine curves. Nevertheless, the transformation (2) clearly preserves the geometry of the classical Koch fractal (self-similar set), and, in addition, satisfies the Equation (7) for it.

## 3. Problem Statement

A wire balanced dipole antenna is considered herein, the arm geometry of which is a second-order Koch-type prefractal. The antenna feed point is located at $(0, 0)$. An example of such a dipole antenna is shown in Figure 3.

The forming prefractal (first-order prefractal) is defined on the interval $[0, 1]$. The arms of the antennas are considered, which are Koch prefractals scaled by a factor of v. That is, the actually forming prefractal is defined on $[0, v]$. Obviously, the same length can be obtained at different scales of v and at different initial positions of the central vertex. Since the base frequency depends on the length of the conductor [31], many antennas of a given type can have the same operating frequency.

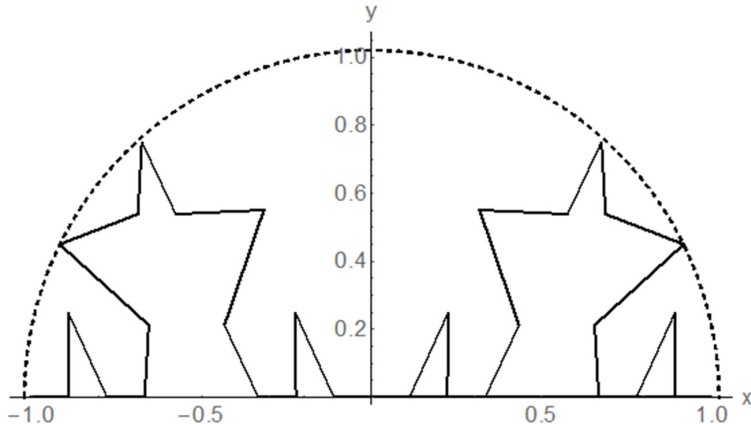

**Figure 3.** Balanced dipole wire antenna with an arm formed by the Koch-type prefractal of the second iteration.

The aim of this work is to design miniature Koch-type fractal antennas operating at a given frequency (2.45 GHz). Moreover, two subtasks can be distinguished. The first subtask is to obtain the smallest antenna of this type. The second subtask is to design a minimum size antenna with a given matching. By antenna size, the minimum radius of a circle describing a dipole is implied (see Figure 3).

For regression analysis at a fixed scale, 440 antennas were calculated in the FEKO program, in which the central vertex of the generator curve (prefractal of the first order) varies within the following limits: The $x$ coordinate changes from 0.25 to 0.75 with a step of 0.025, and the $y$ coordinate changes from 0.25 to 0.8, also with a step of 0.025. The diameter of the antenna wire is 0.4 mm.

## 4. Base Frequency

The dependence of the base frequency on the length of the conductor was analyzed. To perform the analysis, the resonances at various scales v were calculated. The results are presented in Figure 4. The graphs show the dependence of $f_1$ on L. Values of $f_1$ for v = 0.012 m and v = 0.028 m are indicated in black. For v = 0.016 m, the values are in blue; for v = 0.020 m, the values are in red; and for v = 0.024 m, the values are in green.

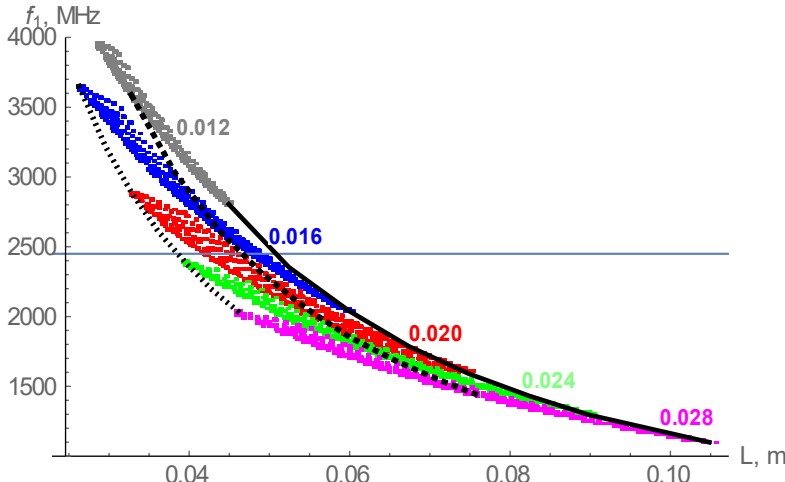

**Figure 4.** Dependence of the base frequency $f_1$ on the length of the arm's wire for various scales. The wire diameter is 0.4 mm. The blue color indicates the calculation results of 440 antennas with various central vertices for v = 0.016 m, etc. The black curves show the dependences at a continuous change in scale for a specific Koch prefractal: the solid line corresponds to the antenna with a central vertex at (0.725, 0.775); the dashed line is for (0.576, 0.538); and the dotted line is for (0.475, 0.25).

Figure 4 shows the points for the scales of various antennas with step $\Delta v = 0.002$ m. Apparently, with a decrease in the step $\Delta v$, the points for identical antennas will continuously change their coordinates from a smaller scale to a larger scale. Examples of such a change for three antennas are shown in Figure 4: the antenna dipole with a central vertex (0.576, 0.538) is shown by a dashed line. It should be noted that the length of the Koch prefractal of the $n$th iteration is calculated as $L = S^n$, where $S = 4/3$ (the value of $S$ is determined from Equation (7)). For the Koch prefractals, there is no explicit formula for calculating $L$; Equation (7) can be used only for estimating the values of $S$.

Note that the upper threshold for computing the base frequency is 4 GHz. With a scale of $v = 0.012$ m, some of the antennas have base frequencies above 4 GHz, so these antennas were not included in our consideration. Thus, a total of 2041 antennas were considered: 281 antennas were obtained for $v = 0.012$ m, whereas for each of the remaining four scales, 440 antennas were obtained.

Following [33], a regression analysis of the set of points in Figure 4 was conducted separately at each scale according to the formula

$$\hat{f}_1 = k_1 \, e^{-k_2 \, L} \tag{8}$$

where $k_1$, $k_2$ are the desired parameters of the model, which are sought by the least squares method. Here, $f_1$ is measured in MHz, and $L$ is measured in meters. The obtained values for the coefficients of the model are given in Table 1.

**Table 1.** Coefficients of regression model (1) for the base frequency.

| Scale, m | $k_1$ | $k_2$ |
|---|---|---|
| 0.012 | 7329.63 | −21.5032 |
| 0.016 | 5824.25 | −17.8957 |
| 0.020 | 4633.42 | −14.6327 |
| 0.024 | 3863.48 | −12.3748 |
| 0.028 | 3306.20 | −10.7503 |

In the next step, a regression model was built by adding a scale to its parameters. In this case, the approach proposed in [33] was used. The following formula was obtained:

$$\hat{f}_1 = \frac{1}{0.000007451 + 0.01048v} \cdot \mathrm{Exp}\left[-\frac{1}{0.00993 + 2.95v}L\right] \tag{9}$$

The relative error for model (2) was calculated by the following formula:

$$\delta = \frac{1}{n}\sum_{i=1}^{n}\left|\frac{Y_i - \hat{Y}(X_i)}{Y_i}\right|100\ \%, \tag{10}$$

which is equal to $\delta \approx 1.22$ %.

## 5. Dipole Antenna Reflection Coefficient

In this section, the behavior of the reflection coefficient $\hat{S}_{11}$ with change in the conductor length is studied. This study was also carried out for various scales. The results are shown in Figure 5. It can be noted that for a fixed scale, long antennas have a matching which is worse than that of short antennas.

Regression analysis of a set of points was conducted separately for each scale using the following formula:

$$\hat{S}_{11} = \frac{k_3}{k_4 - L}, \tag{11}$$

where $k_3$, $k_4$ are unknown parameters of the model. The parameters were also found by the least squares method. The obtained values for the coefficients of the model are given in Table 2.

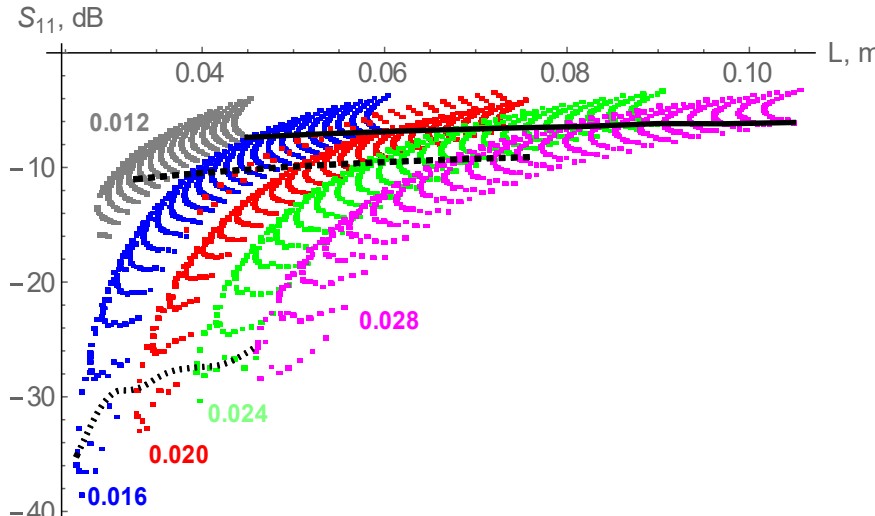

**Figure 5.** The dependence of the reflection coefficient $S_{11}$ on the length of the arm's wire for various scales. The wire diameter is 0.4 mm. The blue color indicates the calculation results of 440 antennas with various central vertices for v = 0.016 m, etc. The black curves show the dependences at a continuous change in scale for a specific Koch prefractal: the solid line corresponds to the antenna with a central vertex at (0.725, 0.775); the dashed line is for (0.576, 0.538); and the dotted line is for (0.475, 0.25).

**Table 2.** Coefficients of regression model (4) for $\hat{S}_{11}$ at the base resonance.

| Scale, m | $k_3$ | $k_4$ |
| --- | --- | --- |
| 0.012 | 0.162 | 0.017 |
| 0.016 | 0.227 | 0.020 |
| 0.020 | 0.271 | 0.024 |
| 0.024 | 0.327 | 0.028 |
| 0.028 | 0.372 | 0.032 |

Analysis of Figure 5 indicates that as $L$ increases, antenna matching worsens. In other words, the more the wire fills the space (the closer the adjacent antenna elements are located to each other), the worse the dipole matching. In Figure 5, the solid black line corresponds to an antenna with a long length, the dashed line corresponds to an antenna with an average length, and the dotted line corresponds to an antenna with a minimum arm length. Thus, the following can be concluded:

$$S_{11}(L_1) < S_{11}(L_2), \quad \text{for } L_1 < L_2. \tag{12}$$

Comprehension of condition (12) leads to the logical conclusion that minimizing the antenna leads to a deterioration in its matching.

The solid black curve corresponds to a normalized (at v = 1) length of 1.64; $S_{11}$ varies from $-7.38$ to $-6.07$ with an increase in scale from 0.012 to 0.028 (with an increase in length from 4.5 to 10.5 cm).

At a continuous change in scale, antenna matching changes continuously.

Just as in the previous paragraph, a regression model for any scale was constructed. The following regression model was obtained:

$$\hat{S}_{11} = \frac{0.012 + 12.986v}{0.00495 + 0.973v - L}, \tag{13}$$

with relative error obtained by formula (13), equal to δ ≈ 12.59 %.

Note that the resulting formula is less accurate than a similar formula for the base frequency. This is also evident from comparison of the graphs in Figures 2 and 3.

## 6. Results

Two examples are considered here. The first example is related to obtaining the smallest antenna. The second example is related to designing a minimum size antenna with a given matching.

### 6.1. Maximum Antenna Miniaturization

Modeling the antenna of minimum radius is considered next. The isoline $\hat{f}_1 = 2450$ MHz was constructed (in Figure 4, it is shown by the curved line). Around the contour, a region $f_1 \in (2.4\,\text{GHz}, 2.5\,\text{GHz})$ is highlighted. The range of allowable values for $L$ is shown in yellow. Thus, the allowable length of the arm length varies from $L = v$ to the maximum possible wire length at a given scale ($L = 3.64\,v$). The maximum value of $L$ is determined from the condition that the antenna is located inside a circle of radius v.

From analysis of Figure 6, it follows that the curve $f_1 = 2450$ MHz crosses the straight line $L = 3.64v$ at v = 0.0138 m. This scale will correspond to the minimum antenna at which the base frequency $\hat{f} = 2450$ MHz is reached. The point of intersection for $L$ is around 0.05 m.

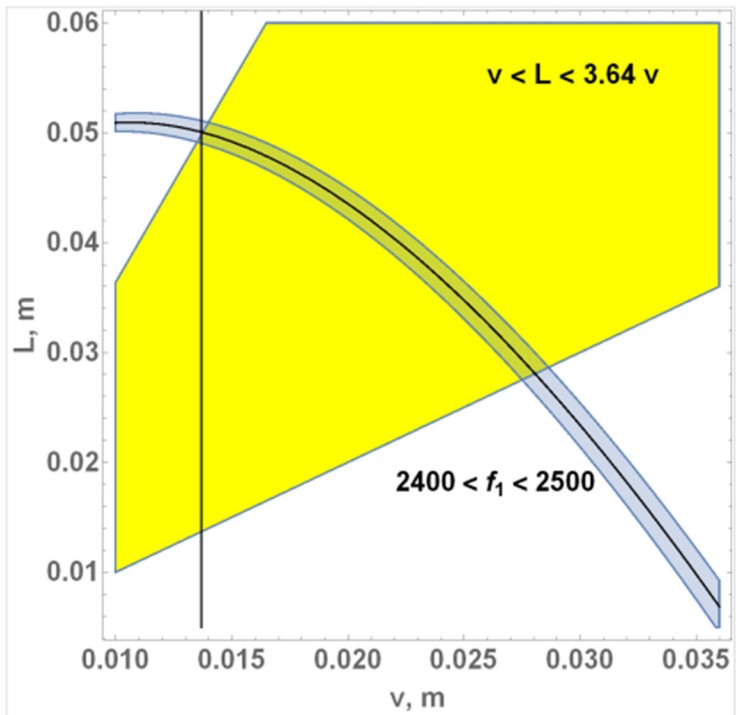

**Figure 6.** Wi-Fi frequency isolines and allowable areas for $L$ and v.

The antenna at the found scale is now considered, with length equal to $L = 0.0497$ m. Such an antenna can be obtained with a forming prefractal having a central vertex at ($0.675 \times 0.0138$ m, $0.75 \times 0.0138$ m). This antenna reaches resonance at the frequency of $\hat{f} = 2.47$ GHz with reflection coefficient $\hat{S}_{11} = -7.2$ dB.

It can be noted that the resulting antenna matching is not the most optimal. One possible further improvement may be rotation of a dipole arm [37,38]. However, simulating an antenna with lower reflectance with no rotation of the arm was attempted. To do this, the dependence of $\hat{S}_{11}$ on the length $L$ is considered.

### 6.2. Designing a Minimum Antenna with a Specified Reflection Coefficient

Now the antenna is modelled, taking into account formula (5). The values of $f_1 = 2450$ MHz and $\hat{S}_{11} = -15$ dB were set. The system of Equations (2) and (5) was solved with respect to the base frequency and the scale. The scale was rounded to v = 0.02 m to obtain $L = 0.043$ m. In this case,

the coordinates of the central vertex of the forming prefractal are (0.625 × 0.02 m, 0.375 × 0.02 m). Based on the simulated parameters, an antenna was fabricated and is shown in Figure 7.

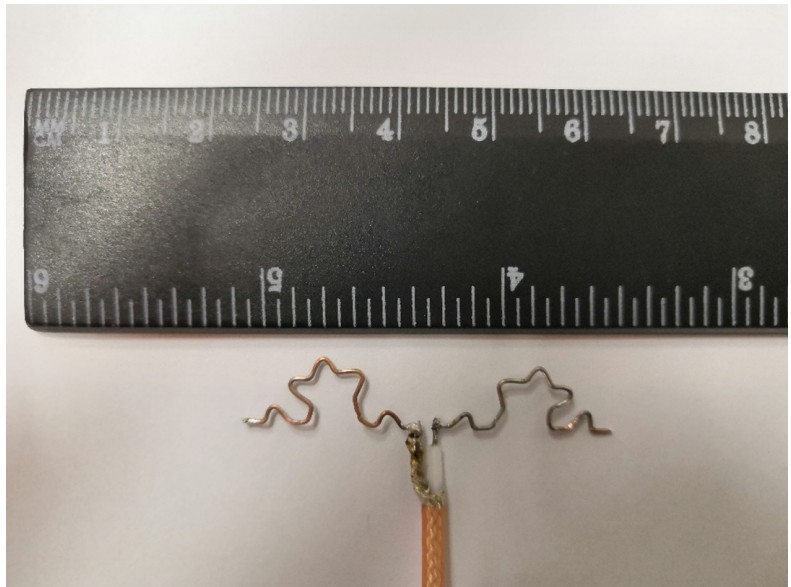

**Figure 7.** The fabricated antenna with central vertex (0.625, 0.375) for v = 0.02 m.

The electrical characteristics of the resulting antenna were calculated. The resulting antenna has reflection coefficient $\hat{S}_{11}$ = −16 dB at the base frequency $f_1$ = 2430 MHz. The results of the calculation of the reflection coefficient of this antenna and of the antenna obtained earlier are shown in Figure 8. An improvement in antenna matching with increasing scale can be observed. By increasing the scale from v = 0.0138 m to v = 0.02 m, the antenna matching can be improved.

In addition, in Figure 8, the red dotted line shows the reflection coefficient for the prototype of the second "optimal" antenna for v = 0.02 m.

Images of the arms of the obtained antennas are presented in Figure 9. Also, to compare the sizes of the antennas, their radii are depicted. The black color indicates the antenna with scale v = 0.02 m, and the blue color indicates an antenna with v = 0.0138 m.

The parameters of a wire dipole based on the Koch curve (a special case of a Koch-type dipole) are presented; like the antennas obtained above, it has a wire diameter of 0.4 mm. The required frequency of 2.44 GHz for this dipole is achieved at a radius of 2.25 cm. This value is greater by 8.7 mm than the radius of the first antenna and greater by 2.5 mm than the radius of the second antenna. The matching of the Koch dipole is better ($S_{11}$ = −21 dB) in comparison with the matching of the obtained optimal antennas. It can also be noted that the matching of the obtained "minimized" antennas can be somewhat improved by applying iterative methods [39].

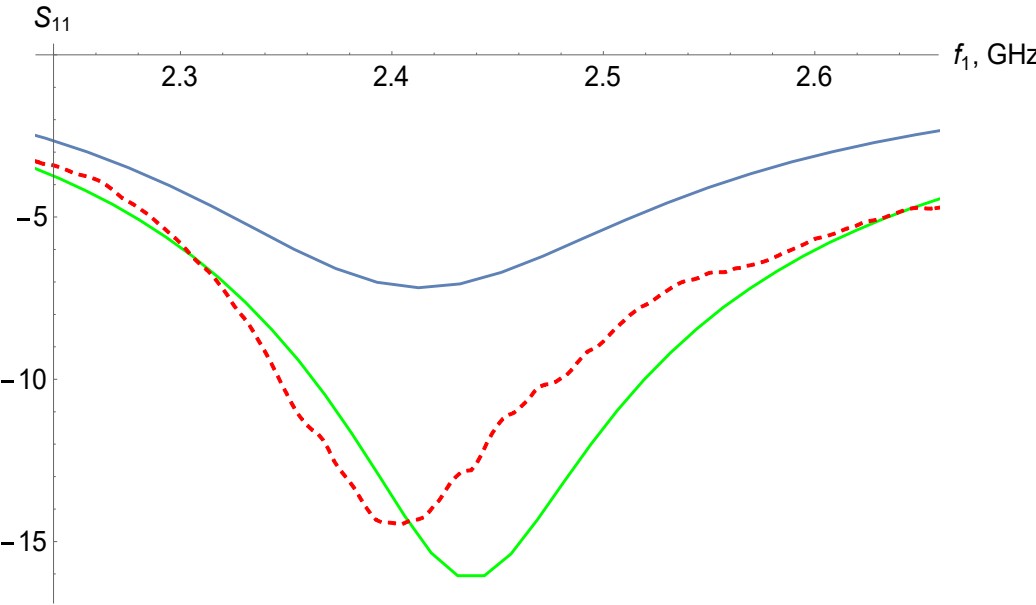

**Figure 8.** The dependence of the reflection coefficient $S_{11}$ on the frequency of the two antennas. The blue graph corresponds to the smallest antenna, the green graph corresponds to the simulated second antenna, and the red graph corresponds to the fabricated second antenna.

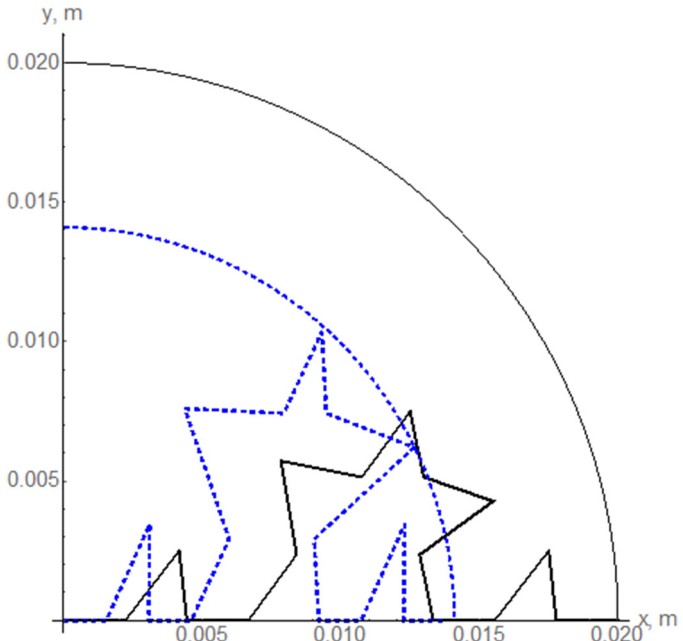

**Figure 9.** The arms and envelopes of the designed antennas. The blue color corresponds to the smallest antenna with $S_{11} = -7$ dB (radius is 14.1 mm), while the black color corresponds to the miniature antenna with $S_{11} = -15$ dB (radius is 20 mm).

It can be noted that the antennas obtained at different scales of v have linear polarization and practically the same radiation pattern and power gain. Figure 10 shows the gain for the obtained optimal antennas at frequency equal to 2430 MHz.

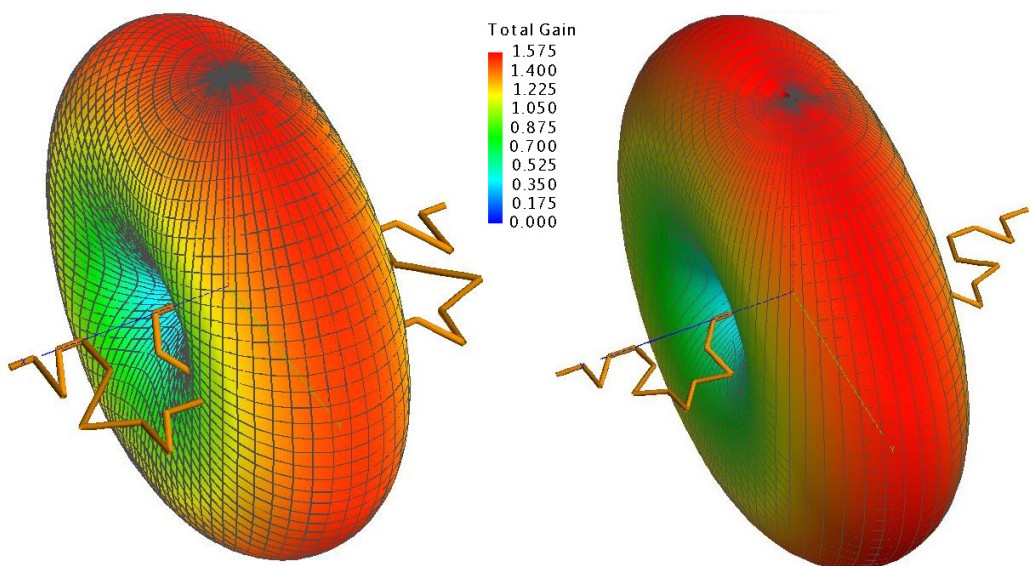

**Figure 10.** Radiation patterns of the obtained antennas at frequency equal to 2430 MHz. Left picture: gain for the antenna with radius equal to 14.1 mm. Right picture: gain for the antenna with radius equal to 20 mm.

## 7. Conclusions

Wire dipole antennas with arm having a Koch-type geometry of the second iteration were considered herein. Two-parameter regression models were constructed for the base frequency and the reflection coefficient at the base frequency for the dipole. A single-band antenna of minimum size (radius of 14.1 mm) was designed. The reflection coefficient of this antenna was equal to −7.2 dB.

In addition, a second antenna with a radius of 2 cm and a reflection coefficient of −16 dB was designed. Both of the proposed antennas were modeled using algorithms based on regression models. The first algorithm allows for designing the minimum size antenna from a given family. The second algorithm gives the minimum size antenna with a given level of matching.

It can be noted that if the goal is to simulate a broadband antenna, then in the present study it is necessary to add a model for the bandwidth [40]. It is also possible to use all models for designing antennas with specified characteristics.

**Author Contributions:** Conceptualization, D.T.; Investigation, D.T. and P.K.; Methodology and Writing Manuscript, D.T. and D.C.; Supervision, D.C. All authors have read and agreed to the published version of the manuscript.

**Funding:** This research received no external funding.

**Acknowledgments:** The work was performed according to the Russian Government Program of Competitive Growth of Kazan Federal University.

**Conflicts of Interest:** The authors declare no conflict of interest.

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
