# Peer review of "Miniaturization of a Koch-Type Fractal Antenna for Wi-Fi Applications"

_fractalfract, doi:10.3390/fractalfract4020025_

Round 1
Reviewer 1 Report
The topic is well described in the scientific/technical literature. Authors try to make the scientific problem from the technical one. Fractal geometry applied to the antenna technology has been used since 1960's.
The problem of frequency tuning of fractal geometry to find the assumed resonant frequency has potential application to the antenna design. But, how authors introduce it to the Koch fractal geometry is not clear. The scaling factor v is related to overall length L of wire used for antenna construction, but each parts of the Koch fractal are in proper proportion each other. How L relates to components of Koch geometry? Do you know if scaling factor for the fractal geometry has completely different meaning-do you distinguish it?
- Author did not show how antenna is designed and next scaled?
- Fig.1 show the dipole wire antenna of the Koch-type which seems to be tilted, similarly like in Fig. 6.
- Why linear antenna has exponential relation (1) of frequency and length?
- Explain the plots in Fig.2 and Fig.3 why they are speeded, not continuous lines, for different v-factors?
- Authors considered "the smallest linear dimensions and minimum antenna for a given reflection coefficient"- do it make sense the minimize antenna which do not realizes correct parameter, reflection coefficient S11~-7dB?
- The antenna basic role is to radiate EM energy to the space, so the most important parameters of it are: radiation pattern, power gain, polarization. What about them in your paper?
- What about frequency bandwidth, is it enough for WiFi system?
- The theory without experiment is not believable!!!
- Let you fabricate your antenna, measure, and compere theory and practice.
- Let you compare your results wit other known from the references
- You reference to the journals. conferences, etc. which are not known, and impossible to check: IJRTE, MMET Int. Conf., IJRECE, ISAPE, App. Math. Model, J. Fundam. Appl. Sci., Proc. EWDTS - abort shortcut, add city, country
Author Response
Thank you for the comments. In the manuscript, our corrections are highlighted in yellow.
- The scaling factor v is related to overall length L of wire used for antenna construction, but each parts of the Koch fractal are in proper proportion each other. How L relates to components of Koch geometry? Do you know if scaling factor for the fractal geometry has completely different meaning-do you distinguish it?
A new text is added into the second paragraph of the section “3. Base frequency”:
“It should be noted that if the length of the Koch prefractal of the nth iteration is calculated as L=(4/3)^n, then for the Koch prefractals there is no explicit formula for calculating L. Therefore, the values of L for each prefractal are calculated by direct addition of the length of the segments of its constituents.”
- Author did not show how antenna is designed and next scaled?
First paragraph in the section “2. Problem statement”:
“A wire balanced dipole antenna is considered, the arm geometry of which is a second-order Koch-type prefractal. The antenna feed point is located at (0, 0). An example of such a dipole antenna is shown in Fig. 1. Note that the Koch-type fractal differs from the Koch fractal only by the position of the central vertex.”
- Fig.1 show the dipole wire antenna of the Koch-type which seems to be tilted, similarly like in Fig. 6.
No, these antennas are just alike.
- Why linear antenna has exponential relation (1) of frequency and length?
Indeed, for an ordinary half-wave dipole, the relationship between the wavelength at the resonant frequency and the dipole length is linear. However, for a wire dipole with a complex geometry of the arm, formula (1) is more accurate [17].
[17] Tumakov, D.N.; Abgaryan, G.V.; Chickrin, D.E.; Kokunin, P.A. Modeling of the Koch-type wire dipole, Applied Mathematical Modelling (Appl. Math. Model.) 2017, 51, 341-360.
- Explain the plots in Fig.2 and Fig.3 why they are speeded, not continuous lines, for different v-factors?
The following text is added: “Fig. 2 shows the points for the scales of various antennas with the step ∆v = 0.002 m. Apparently, with a decrease in the step ∆v, the points for identical antennas will continuously change their coordinates from a smaller scale to a larger scale. An example of such a change for an antenna with a central vertex (0.576, 0.538) is shown in Fig. 2 (magenta curve).”
For Fig. 3, all points will also be continuously connected.
- Authors considered "the smallest linear dimensions and minimum antenna for a given reflection coefficient"- do it make sense the minimize antenna which do not realizes correct parameter, reflection coefficient S11~-7dB?
Our task was to build a minimum antenna without specifying a condition for the reflection coefficient. I think that in certain applications such antenna can be used.
- The antenna basic role is to radiate EM energy to the space, so the most important parameters of it are: radiation pattern, power gain, polarization. What about them in your paper?
The following text was added: “It can be noted that the antennas obtained at different scales of v have linear polarization and practically the same radiation pattern and power gain.”
- What about frequency bandwidth, is it enough for Wi-Fi system?
Antennas with the obtained values for bandwidth are quite applicable. In the Conclusion, it is said: “It can be noted that if the goal is to simulate a broadband antenna, then in the present study it is necessary to add a model for the bandwidth [28]. It is also possible to use all models for designing antennas with specified characteristics.”
- The theory without experiment is not believable!!!
- Let you fabricate your antenna, measure, and compere theory and practice.
Surely, if we had a designed and fabricated antenna in hand, then the article would be definitely improved. However, I would like to make an argument that, in our opinion, modern software products such as FEKO, HFSS, etc., calculate wire antennas with a very high accuracy.
Also in the near future we will not be able to go to work due to quarantine.
- Let you compare your results with other known from the references
Unfortunately, our present work is specific in nature, and it is not possible to compare results for this type of antenna versus other published results. In the overwhelming majority of cases, the authors of previous works design a specific antenna and obtain its geometry by a method, which is not clearly disclosed and, therefore, remains unavailable to us.
- You reference to the journals. conferences, etc. which are not known, and impossible to check: IJRTE, MMET Int. Conf., IJRECE, ISAPE, App. Math. Model, J. Fundam. Appl. Sci., Proc. EWDTS - abort shortcut, add city, country
The references are corrected.
Reviewer 2 Report
Wire dipole antennas whose arm has a Koch-type geometry of the second iteration are presented.
It is recommended to compare the performance with a conventional koch wire dipole antenna as well.
What is the miniaturization factor?
Author Response
It is recommended to compare the performance with a conventional Koch wire dipole antenna as well.
A comparison of the obtained antennas versus a dipole having the geometry of the second iteration of the Koch curve is added. “The parameters of a wire dipole based on the Koch curve (a special case of a Koch-type dipole) are presented, which, like the antennas obtained above, has a wire diameter of 0.4 mm. The required frequency of 2.44 GHz for this dipole will be achieved at a radius of 2.25 cm, which is greater by 8.7 and 2.5 mm than radii of the obtained antennas. The matching of the Koch dipole will be better (S_11 = –21 dB), which is consistent with the results presented in Fig. 3. It can also be noted that the matching of the obtained “minimized” antennas can be somewhat improved by applying iterative methods [27].”
What is the miniaturization factor?
We did not use such a term in the present work. But perhaps you are talking about by how many times the antenna can be reduced in comparison with an ordinary half-wave dipole. In our case, this may be the ratio of the wire length L to the scale v.
Reviewer 3 Report
The work refers to design, development and optimization of the Koch fractal based miniaturization of radiating system for WiFi applications. The manuscript is well written in 6 Sections with excellent math background and clear description of the methodology. Though the concept of minimum antenna is rather tricky, needs justification and clarity.
THe key and appreciable part of the work is the regression model of the antenna.
The EM simulator and its features are well utilized for tuning and optimizing the antenna physical parameters with the objective of required radiation and electrical characteristics.
I suggest the following revisions.
- Thorough revision of the language.
- Paper organization in the introduction
- The discussion in results Section need more inputs and inference on study made.
- Avoid using "we" which is extensively used throughout the paper.
- Suggested few references to be cited from this journal.
- Fabrication results are essential to validate the results. However, it is left to the authors perception of the problem.
Author Response
Thorough revision of the language.
The text is corrected by a native speaker.
The discussion in results Section need more inputs and inference on study made.
At the end of the section, the following paragraph is added:
“The parameters of a wire dipole based on the Koch curve (a special case of a Koch-type dipole) are presented, which, like the antennas obtained above, has a wire diameter of 0.4 mm. The required frequency of 2.44 GHz for this dipole will be achieved at a radius of 2.25 cm, which is greater by 8.7 and 2.5 mm than radii of the obtained antennas. The matching of the Koch dipole will be better (S_11 = –21 dB), which is consistent with the results presented in Fig. 3. It can also be noted that the matching of the obtained “minimized” antennas can be somewhat improved by applying iterative methods [27].”
Avoid using "we" which is extensively used throughout the paper.
This comment may be challenged, but we agreed and got rid of “we”.
Suggested few references to be cited from this journal.
We were ready to do it right away at the very beginning of work on the manuscript. However, we have just looked through all the existing journal issues once again and could not find even a single work in the field of fractal electrodynamics, as well as in the field of fractals to complement the Introduction in a natural manner (due to a young age of the journal).
Fabrication results are essential to validate the results. However, it is left to the authors perception of the problem.
Surely, if we had a designed and fabricated antenna in hand, then the article would be definitely improved. However, I would like to make an argument that, in our opinion, modern software products such as FEKO, HFSS, etc., calculate wire antennas with a very high accuracy.
Round 2
Reviewer 1 Report
The authors did not used the reviewer suggestions to make the paper acceptable for publication.
- Fig. 1 presents tilted geometry, similar to the Koch fractal.
- Authors did not explain their knowledge bout fractal scaling factor and relation of the fractal geometry to the scaling component v. The scaling of the whole length of wire used for antenna fabrication do nit show how fractal should be modified. So, design procedure of the antenna is not clarified!
- The write: ".. the values of L for each prefractal are calculated by direct addition of the length the segment of its constitutes." ???? After that modification it will not satisfy the Koch fractal geometry.
- Plots on Fig. 2, and Fig. 3 are not corrected, and present unequivocal results (e.g. in Fig. 3, for proper value of L you have two or more different values of S), the should be in form of continuous lines, not spreaded points!
- The relationship between L and f in expressions (1) and (2) is not clearly understand, why it is unlinear?
- "... Our task was to build minimum antenna without specifying a condition..." - it is not professional approach!
- Theoretical results without experimental or comparison with other method are not believable and useful.
- Authors present wrong statement: "... modern software products ... calculate wire antennas with very high accuracy."-it is not that, especially for unknown antenna structures.
- Antenna parameters, like power gain, radiation patterns, should be determined - not guessing!
- Acronyms in References, especially of some "exotic" journals or conferences are not well recognized, so it should be described more detail.
Author Response
- 3 presents tilted geometry, similar to the Koch fractal.
Yes, you are right. Figure 3 shows a Koch-type curve similar to the classical Koch curve. The iterating curve has a lateral tilt. An example of the first three iterations of such a curve is shown in Fig. 2.
- Authors did not explain their knowledge bout fractal scaling factor and relation of the fractal geometry to the scaling component v. The scaling of the whole length of wire used for antenna fabrication do nit show how fractal should be modified. So, design procedure of the antenna is not clarified!
Perhaps, in our present paper the ordinary scale v and fractal dimension were somewhat confused. We accepted this remark and made a decision to describe in detail the construction of "our" Koch-type curves. A new section (Section 2) was added.
- The write: ".. the values of L for each prefractal are calculated by direct addition of the length the segment of its constitutes." ???? After that modification it will not satisfy the Koch fractal geometry.
Koch-type curves are self-affine, and in general, formulas of the form (7) for determining the dimension are not satisfied. Consequently, we can only analytically evaluate the length of the curve, and for an accurate determination, it is required to perform direct calculations.
- Plots on Fig. 2, and Fig. 3 are not corrected, and present unequivocal results (e.g. in Fig. 3, for proper value of L you have two or more different values of S), the should be in form of continuous lines, not spreaded points!
- Explanations were added to the figures. Yes, with a continuous change in the scale v, both the frequency and the reflection coefficient will change continuously.
- The relationship between L and f in expressions (1) and (2) is not clearly understand, why it is unlinear?
Indeed, the relationship between the wavelength at the resonant frequency and the arm length can be obtained as a linear relationship. As shown in the work by Tumakov D.N. et al. "Modeling of the Koch-type wire dipole", the connection between f and L is better in the form given in that work.
- "... Our task was to build minimum antenna without specifying a condition..." - it is not professional approach!
- It was corrected.
- Theoretical results without experimental or comparison with other method are not believable and useful.
- A new prototype of the antenna was fabricated. The values were measured.
- Authors present wrong statement: "... modern software products ... calculate wire antennas with very high accuracy."-it is not that, especially for unknown antenna structures.
- We removed this part, although we believe that the accuracy of calculation of wire antennas is significantly higher than the accuracy of calculation of microstrip antennas.
- Antenna parameters, like power gain, radiation patterns, should be determined - not guessing!
- Calculated values of gain were added to Figure 10.
- Acronyms in References, especially of some "exotic" journals or conferences are not well recognized, so it should be described more detail.
- All journals are written in full, abbreviations are left in brackets.
Reviewer 3 Report
Some of the revisions suggested are incorporated.
1. I really do not understand the stand of the authors to challenge the suggestions to avoid the usage of "We" which is a must practice in research papers. However, changes are made.
2. What does the included statement mean? It appears these are pertinent int he results while the statement mentions of action to be taken in future. A minor revision for clarity and better readability is appreciable.
"The required frequency of 2.44 GHz for this dipole will be achieved at a radius of 2.25 cm, which is greater by 8.7 and 2.5 mm than radii of the obtained antennas. The matching of the Koch dipole will be better (S_11 = –21 dB), which is consistent with the results presented in Fig. 3".
Also, suggested to check the clarity of the statement. If possible simplify.
It should be noted that if the length of the Koch prefractal of the nth iteration is calculated as ? = (4/3)? , then for the Koch prefractals
86 there is no explicit formula for calculating L.
3. Though the modern EM tools are efficient, the validation with the prototypes is always suggested. However, many other things are associated and hence left to the possibilities and abilities of the authors (citing their previous works mentioned in the refs (#17, #25) where there is no such traces of prototyping).
4. Regarding the citation of the works from this journal: It is a very quick statement made by the authors. The journal is specific to fractals which is unique of its own kind and has some issues published already with very much relevancy to this work. With good indexing and experience of the publisher, I think statement can no longer be a valid one. Strongly suggested to review this if you agree.
Out of 28 Refs, 9 are from the author itself. Generally an authors latest publication in the field should consolidate his previous. Hence citing his latest would be best ethical practice.
Author Response
- What does the included statement mean? It appears these are pertinent int he results while the statement mentions of action to be taken in future. A minor revision for clarity and better readability is appreciable.
It was corrected.
- Though the modern EM tools are efficient, the validation with the prototypes is always suggested. However, many other things are associated and hence left to the possibilities and abilities of the authors (citing their previous works mentioned in the refs (#17, #25) where there is no such traces of prototyping).
A prototype of the antenna was fabricated, and measurements were performed.
- Regarding the citation of the works from this journal: It is a very quick statement made by the authors. The journal is specific to fractals which is unique of its own kind and has some issues published already with very much relevancy to this work. With good indexing and experience of the publisher, I think statement can no longer be a valid one. Strongly suggested to review this if you agree.
We removed one of our sources and added 11 new ones (almost all of the journals were published by MDPI).